# Altered Lung Heat Shock Protein-70 Expression and Severity of Sepsis-Induced Acute Lung Injury in a Chronic Kidney Disease Rat Model

**DOI:** 10.3390/ijms24065641

**Published:** 2023-03-15

**Authors:** Jun-Yeun Cho, Seung-Jung Kim, Chang-Gok Woo, Soon-Kil Kwon, Kang-Hyeon Choe, Eung-Gook Kim, Yoon-Mi Shin

**Affiliations:** 1Department of Internal Medicine, Chungbuk National University Hospital, Chungbuk National University College of Medicine, Cheongju 28644, Republic of Korea; ok_kaist2115@hanmail.net (J.-Y.C.);; 2Department of Pathology, Chungbuk National University Hospital, Chungbuk National University College of Medicine, Cheongju 28644, Republic of Korea; 3Department of Biochemical and Medical Research Center, Chungbuk National University College of Medicine, Cheongju 28644, Republic of Korea

**Keywords:** sepsis, acute lung injury, chronic kidney disease, heat shock protein-70

## Abstract

Enhanced heat shock protein-70 (HSP-70) expression in the lungs is associated with attenuated acute lung injury (ALI) in a sepsis model. Chronic kidney disease (CKD) significantly contributes to the poor prognosis of patients with sepsis. This study examined the relationship between sepsis-induced ALI severity and altered lung HSP-70 expression in CKD. Experimental rats underwent a sham operation (control group) or 5/6 nephrectomy (CKD group). Sepsis was induced with cecal ligation and puncture (CLP). Laboratory tests and lung harvest were performed in the control group (without CLP and after 3, 12, 24, and 72 h of CLP) and in the CKD group (without CLP and after 72 h of CLP). ALI was the most severe after 12 h of sepsis. The mean lung injury score at 72 h after sepsis was significantly higher in the CKD group than in the control group (4.38 versus 3.30, *p* < 0.01). Nonetheless, enhanced lung HSP-70 expression was not observed in the CKD group. This study shows that altered lung HSP-70 expression is associated with the worsening of sepsis-induced ALI in patients with CKD. Enhancing lung HSP-70 is a novel treatment target for patients with CKD and sepsis-induced ALI.

## 1. Introduction

### 1.1. Sepsis-Induced Acute Lung Injury

Acute lung injury (ALI) is a hypoxemic clinical syndrome characterized by bilateral and diffuse lung infiltration [1]. Various pulmonary or non-pulmonary conditions can cause ALI. Sepsis is defined as a life-threatening organ dysfunction caused by a dysregulated host response to infection [2]. Sepsis can frequently induce ALI [3]. Previous studies have reported that approximately 6% of patients with sepsis developed ALI [4,5]. Upregulated inflammatory cytokines (such as tumor necrosis factor (TNF)-α, interleukin (IL)-1β, IL-6, IL-8, and IL-18) are the key mediators of the pathophysiology of sepsis-induced ALI and biomarkers for predicting clinical outcomes [6]. Among them, IL-6, which is mainly produced by the innate immune system, is one of the essential cytokines released during the acute phase of ALI.

### 1.2. Role of Heat Shock Protein in Acute Lung Injury

The heat shock response (HSR), classically described as a response to thermal stress, is a biological defense mechanism against a variety of cellular and tissue injuries [7]. The exact mechanism has not been identified; however, all organisms have the intrinsic function of inducing HSR. Heat shock proteins (HSPs), which are rapidly expressed through HSR, consist of different types of proteins according to molecular weight. Cellular stress in the lungs (such as lipopolysaccharide, free radicals, thermal injury, and hypoxia) induced HSP expression in both alveolar macrophages and epithelial cells, which seems to ameliorate ALI and improve survival [8].

### 1.3. Chronic Kidney Disease (CKD) and Lung Disorder

CKD, a heterogeneous disorder that affects the structure and function of the kidneys, is generally associated with old age, diabetes, and hypertension [9]. CKD prevalence is increasing worldwide [10]. Along with CKD progression, disease-related complications (including hypertension, cardiovascular disease, anemia, and mineral bone disorder) decisively impact the prognosis of patients with CKD [11]. Moreover, CKD prognosis worsens when accompanied by acute medical problems such as sepsis [12].

The kidney–lung cross-talk, an interplay between the lung and the kidney, is well documented [13]. Hypoxia, hypercapnia, and inflammatory mediators induced by ALI, directly or indirectly affect the kidneys, leading to acute kidney injury (AKI). Conversely, AKI results in lung inflammation owing to the direct effect of uremic toxins and inflammatory mediators secreted by tubular damage. In addition, salt and water retention can lead to lung edema and exacerbate ALI.

Patients with CKD are susceptible to ALI development. Nemmar et al., reported that lung inflammation and fibrosis were more severe in an adenine-treated CKD mouse model than in the controls due to lung oxidative stress, DNA damage, and apoptosis during CKD induction [14]. Moreover, Mukai et al., reported a correlation between CKD and impaired lung function, which is believed to be due to several factors such as fluid overload, metabolic and endocrine abnormalities, and decreased respiratory muscles caused by CKD [15].

### 1.4. Objective of the Study

In CKD scenarios, detrimental environments (such as the presence of uremic toxins, inflammatory mediators, reactive oxygen species, and infection) may affect HSP normal expression [16]. Marzec et al., reported that HSP-72 expression in blood monocytes was lower in patients with pre-dialysis CKD and hemodialysis than in normal individuals [17]. Musial et al., observed no significant differences in serum HSP-70 levels between pediatric patients with CKD and healthy individuals [18]. Lin et al., found that rats subjected to subtotal nephrectomy showed lower HSP-27 expression in the aortic wall than controls [19]. The differences in these results can be due to the uremic toxin exposure period.

Currently, it is believed that various HSPs would play an essential role in the defense mechanism against sepsis. However, it is unclear whether normally expressed HSP can alleviate sepsis-induced ALI in a CKD environment where altered HSP expression is expected. In this study, we compared the severity of sepsis-induced ALI and HSP-70 expression in lung tissue between control and CKD rat models. This study aimed to elucidate whether altered lung HSP-70 expression is associated with aggravated sepsis-induced ALI in CKD.

## 2. Results

### 2.1. Clinical Parameters

The clinical parameters of the control and CKD groups were compared in detail (Table 1). Significant body weight loss was observed in the control + sepsis 72 h group compared with the control group (mean ± standard deviation, 451.2 ± 51.0 g vs. 520.0 ± 21.3 g, *p* < 0.05). Urine volume during 24 h did not differ between the control and control + sepsis 72 h groups; however, it was significantly decreased in the CKD + sepsis 72 h group compared with the CKD group (23.4 ± 8.8 mL vs. 30.5 ± 5.2 mL, *p* < 0.01). Urine pH was significantly increased after sepsis induction (control vs. control + sepsis at 72 h (7.43 ± 0.55 vs. 9.16 ± 0.12, *p* < 0.01), CKD vs. CKD + sepsis 72 h (7.56 ± 0.28 vs. 8.67 ± 0.27, *p* < 0.01)).

Serum creatinine, blood urea nitrogen (BUN), and creatinine clearance levels did not differ between the control and control + sepsis 72 h groups. BUN level was significantly decreased in the CKD + sepsis 72 h group compared with the CKD group (37.4 ± 4.5 mg/dL vs. 53.6 ± 14.4 mg/dL, *p* < 0.05). Along with a decrease in BUN, creatinine clearance was significantly elevated in the CKD + sepsis 72 h group compared with the CKD group (0.21 ± 0.03 mL/min/100 g vs. 0.15 ± 0.04 mL/min/100 g, *p* < 0.01). However, serum creatinine did not differ between the CKD and CKD + sepsis at 72 h groups. Serum creatinine, BUN, and creatinine clearance levels were significantly elevated in the CKD group compared with the control group (serum creatinine, 0.90 ± 0.26 mg/dL vs. 0.40 ± 0.03 mg/dL, *p* < 0.01; BUN, 53.6 ± 14.4 vs. 22.9 ± 3.9, *p* < 0.01; creatinine clearance, 0.37 ± 0.06 mL/min/100 g vs. 0.15 ± 0.04 mL/min/100 g, *p* < 0.01).

Serum albumin level was significantly decreased in the control + sepsis 72 h group compared with the control group (2.30 ± 0.2 mg/dL vs. 1.78 ± 0.2 mg/dL, *p* < 0.01), whereas it did not differ between the CKD and CKD + sepsis 72 h groups. Regardless of sepsis induction, serum C-reactive protein (CRP) levels did not differ between the control and CKD groups.

### 2.2. Severity of Acute Lung Injury

The photomicrographs of the harvested lung stained with hematoxylin–eosin and changes in lung injury scores according to the course of sepsis are shown in Figure 1 (magnification, 200×). Calculated lung injury scores were as follows: control, 1.00 ± 0.27; the scores for control + sepsis at 3 h, 12 h, 24 h, and 72 h were 1.69 ± 0.18, 4.79 ± 0.02, 3.23 ± 0.13, and 3.30 ± 0.14, respectively; CKD, 1.00 ± 0.35; and CKD + sepsis 72 h, 4.38 ± 0.01. The histological evidence of ALI (that is, the interstitial thickening and infiltration of neutrophils in the interstitial and alveolar spaces) was observed after sepsis induction in both the control and CKD groups.

The lung histological changes were most prominent in the control + sepsis 12 h group (the lung injury scores of the control and control + sepsis 12 h group were 1.00 ± 0.27 and 4.79 ± 0.02, respectively, *p* < 0.01). These changes were alleviated in the control + sepsis 24 h and 72 h groups. The lung injury score was significantly higher in the control + sepsis 72 h group than in the control group (3.30 ± 0.14 vs. 1.00 ± 0.27, *p* < 0.01). Additionally, it was significantly higher in CKD + sepsis 72 h group than in the CKD group (4.38 ± 0.01 vs. 1.00 ± 0.35, *p* < 0.05). The lung injury scores were similar between the control and CKD groups. However, it was significantly higher in the CKD + sepsis 72 h group than in the control + sepsis 72 h group (4.38 ± 0.01 vs. 3.30 ± 0.14, *p* < 0.05).

### 2.3. Lung HSP-70 Expression

Immunohistochemistry for HSP-70 in the harvested lung tissue is shown in Figure 2. The lung HSP-70 expression was most intense in the control + sepsis 12 h group, with a gradual decrease in the control + sepsis 24 h and 72 h groups. The control and control + sepsis 72 h groups had similar HSP-70 expression. A significantly enhanced lung HSP-70 expression was not observed in the CKD + sepsis 72 h group compared with that in the CKD group; however, an increasing trend was observed. Regardless of sepsis induction, no difference in the lung HSP-70 expression was observed between the control and CKD groups.

### 2.4. Serum IL-6 Immunoassay

The immunoassays for serum IL-6 levels are shown in Figure 3. The serum IL-6 density was the highest in the control + sepsis 12 h group. The trends in serum IL-6 changes were similar to those of the lung injury score or lung HSP-70 expression. The serum IL-6 levels did not differ between the control and control + sepsis 72 h groups. However, it was significantly decreased in the CKD + sepsis 72 h group compared with the CKD group.

## 3. Discussion

This study showed that sepsis-induced ALI was aggravated without enhanced lung HSP-70 expression in CKD. The causal relationship between lung HSP-70 expression and sepsis-induced ALI is unclear; however, to the best of our knowledge, this is the first study to focus on the association between sepsis-induced ALI and altered HSP-70 expression in CKD. Our results suggest that the intrapulmonary cytoprotective response against sepsis-induced ALI is not appropriately enhanced in detrimental conditions, such as CKD.

In our study, we successfully induced CKD in experimental animals, as intended. There are several methods for developing CKD experimental animal models, each of which has its own characteristics. The remnant kidney model (5/6 nephrectomized) has the disadvantage of high experimental animal mortality due to surgical procedures. However, it has the advantage of inducing decreased kidney function through renal mass reduction, similar to progressive renal failure in humans [20]. Renal mass reduction can lead to glomerular hypertension and hyper-infiltration, oxidative stress, and inflammation. Eventually, glomerulosclerosis, tubulointerstitial injury, and renal atrophy would lead to end-stage renal disease. The CKD rat model in our study showed relevant features of proteinuria, increased urine volume, and worsened renal function, which is consistent with the results of a previous study [21].

In CKD, various HSPs protect cells against the damage caused by uremic toxins [22,23]. Apoptosis increases with CKD progression, which induces HSP release. Experimental studies have shown that renal HSP-70 expression is increased in response to the severity of renal injury [24,25]. However, HSP expression is decreased in patients with advanced CKD. Marzec et al., showed that diminished monocytic HSP-72 expression was detected in patients before hemodialysis compared with healthy individuals [17]. This contrary phenomenon can be explained as the exhaustion of the adaptive mechanism in the detrimental conditions of progressive CKD. In a study of pediatric patients with CKD, serum HSP-70 concentration was unchanged compared with that in normal individuals [18]. In our study, a prolonged course of uremic exposure may inhibit or reduce the normally upregulated HSP expression. Hence, relatively prolonged periods of uremic toxin exposure (that is, 8 weeks after 5/6 nephrectomy) may suppress the enhancement of lung HSP-70 expression against sepsis-induced ALI.

HSP-70 is one of the most widely studied HSPs. It appears to have protective effects against sepsis by facilitating protein folding and preventing the secretion of inflammatory mediators, although this mechanism remains unclear [26]. HSP-70 showed a protective effect against ALI in an animal sepsis model. Furthermore, HSP-70 gene-deficient mice exhibited increased lung inflammation and systemic levels of TNF-α, IL-6, IL-10, and IL-1β [27]. Lin et al., reported that increased lung HSP-70 expression was associated with the amelioration of hypoxia-induced lung injury in a hypobaric hypoxia-preconditioned rat model [28]. Glutamine, an inducer of HSP-70 expression, showed an improved survival rate in experimental animals with reduced ALI and pro-inflammatory cytokines [29].

We hypothesized that ALI would be most severe after 72 h of sepsis. Park et al., reported that the intrapulmonary concentration of inflammatory mediators peaked on day 3 of acute respiratory distress syndrome (ARDS) [30]. In a mouse model infected with *Pseudomonas*, the intrapulmonary levels of inflammatory cytokines (TNF-α, IL-1β, and IL-6) and neutrophils were the highest on day 3 of infection [31]. Lung injury became prominent 48 h after CLP in experimental studies [32,33]. Therefore, we compared the main parameters between the control and CKD groups at 72 h after CLP.

However, in this study, ALI was most severe at 12 h after sepsis in the control group. This can be due to the relatively mild sepsis in this group. Serum CRP levels were not significantly different between the control and CKD groups, regardless of sepsis induction. In addition, the lung histological findings were not consistent with the severe ALI pattern. We rarely observed hyaline membranes in the alveolar space or proteinaceous edema, which are considered hallmarks of the major histologic features of human ARDS lungs [34]. In experimental animal models, the presence of hyaline membranes or proteinaceous debris in the alveolar space is a relevant pathological piece of evidence of ARDS [35]. In addition, the CLP protocol of our study (that is the single puncture, 70% cecal ligation length, 20-gauge needle size, and volume resuscitation) may have led to a milder septic insult than expected. The CLP experimental model in which peritoneal polymicrobial infection occurs has been widely used to study human sepsis and septic shock [36]. The size of the puncture needle, the number of punctures, the volume of resuscitation, and cecal ligation length are important factors in determining the mortality and septic status of experimental animals [37,38,39].

The strength of sepsis was relatively weak; however, it can be considered that the inflammatory responses resulting from sepsis were appropriately elicited. The trend of lung HSP-70 expression was comparable to that of sepsis-induced ALI, which was similar to the pattern of HSP expression after urea exposure in human neuroblastoma cells [40]. Moreover, the pattern of changes in sepsis-induced ALI was comparable to that in lung HSP-70 expression or serum IL-6 levels.

This study had several limitations. First, we could not assess lung histology, HSP-70 expression, or inflammatory markers at 3 h, 12 h, and 24 h of sepsis in the CKD group. Hence, a clear trend in ALI severity and HSP-70 levels was not evaluated. Second, the intrapulmonary inflammatory markers were not evaluated. Bronchoalveolar lavage was not performed in the experimental animals. Bronchoalveolar lavage fluid analysis can quantitatively evaluate intrapulmonary inflammatory cells and the related mediators. Third, we did not conduct additional experiments to evaluate the causal relationship between sepsis-induced ALI and lung HSP-70 expression. Further studies are needed to address these limitations. In addition, it is necessary to evaluate whether sepsis-induced ALI can be ameliorated by therapeutic candidates in experimental CKD models. This approach is considered essential for treating sepsis-induced ALI, which can be fatal in patients with CKD.

## 4. Materials and Methods

### 4.1. Animals

Male Sprague Dawley rats were obtained from Daehan Biolink (Chungbuk, South Korea) and maintained in a controlled temperature and humidity with adequate food and water supply. The required sample size was calculated using Mead’s resource equation. The rats were randomized into seven groups with five animals in each group: control; control with 3 h, 12 h, 24 h, and 72 h after sepsis (abbreviated as sepsis 3 h, 12 h, 24 h, and 72 h, respectively); CKD; and CKD with 72 h of sepsis (abbreviated as CKD + sepsis 72 h).

### 4.2. Sepsis and Chronic Kidney Disease Model

Cecal ligation and puncture (CLP) and CKD models were developed according to a previously described method [41]. The rats were anesthetized by injecting 50 mg/kg tiletamine plus zolazepam (Zoletil) and 10 mg/kg xylazine (Rompun) into their thigh muscles. A 2-cm incision was made in the intramuscle, fascia, and peritoneum. The cecum was located and exteriorized. The total length of the cecum was measured from the tip of the ascending cecum to the bottom of the descending cecum.

The cecum was ligated at 70% of its total length and perforated with a single puncture midway between the ligation point and the tip of the cecum using a 20-gauge needle. After the needle was removed, fecal extrusion was confirmed. The cecum was reattached, after which the fascia, abdominal musculature, and peritoneal skin were closed using simple running sutures. Immediately after the procedure, saline (5 mL/100 g) was subcutaneously administered for fluid resuscitation [42]. Lung tissue, blood, and urine samples were collected at 3 h, 12 h, 24 h, and 72 h after surgery.

In the CKD group, the lower and upper third of the left kidneys were resected. After a week, the right kidney was removed (5/6 nephrectomy), and a sham operation was performed in the control group. To induce CKD with sepsis, we performed cecal ligation and puncture 8 weeks after CKD induction. After 3 days, lung tissues, blood, and urine samples were collected. The experimental design is illustrated in Figure 4.

### 4.3. Blood and Urine Test

Blood samples were obtained via femoral venous sampling, centrifuged, and stored at −80 °C. Urine samples were collected, and urine volume was measured over 24 h. Creatinine clearance was assessed using a mineral-oil-treated metabolic cage on the day of organ harvest. Body weight and serum glucose, creatinine, and albumin levels were measured using a Nova Stat Profile M Critical Care Analyzer (Nova Biomedical, Waltham, MA, USA). The pH of fresh urine was measured using an Orion 3 Star Plus pH meter (Thermo, Waltham, MA, USA).

### 4.4. Histologic Analysis and Immunohistochemistry

Outer one-half of the left lung was harvested, fixed in an 8% periodate–lysine–paraformaldehyde solution for 8 h at room temperature, stored at 4 °C overnight, and embedded in paraffin. Paraffin blocks were sectioned at 4 µm and stained with hematoxylin and eosin. For immunohistochemistry, the tissue sections were rinsed in xylene to remove paraffin and then rehydrated in a gradient of 70–100% ethanol. Endogenous peroxidase activity was inhibited with 3% H_2_O_2_ at 4 °C for 30 min. The slides treated with normal goat serum (diluted 1:50; Vector Laboratories, Inc., Burlingame, CA, USA) were exposed to the primary antibodies at 4 °C overnight, followed by a biotinylated goat anti-rabbit immunoglobulin G (diluted 1:100; Vector Laboratories, Inc., Burlingame, CA, USA) at room temperature for 30 min. Next, the lung tissue sections were treated with 3,3′-diaminobenzidine substrates, rinsed with xylene, and mounted.

Two pathologists who are specialists in respiratory disorders evaluated the severity of the lung injury by calculating the lung injury score in a blinded manner. Scores were assigned to each of the four parameters (neutrophils in the alveolar space, neutrophils in the interstitial space, proteinaceous debris filling the air spaces, and alveolar septal thickening), and calculated according to the prescribed method [43].

### 4.5. Western Blotting

HSP-70 expressions were measured in the lung tissue through Western blotting using specific antibodies against HSP-70 (diluted 1:1000; Cell Signaling, Danvers, MA, USA), and β-actin (diluted 1:1000; Sigma-Aldrich, St. Louis, MO, USA) was used as the loading control. Proteins were extracted using the Pro-Prep protein extraction solution (Intron, Seoul, South Korea) and assayed spectrophotometrically.

The samples were loaded onto 10% polyacrylamide–sodium dodecyl sulfate mini gels and transferred to polyvinylidene fluoride membranes. The membranes were blocked for 2 h in 0.1% Tris-buffered saline plus Tween 20 (TBS-T) containing 5% non-fat dry milk and treated with primary antibodies against HSP-70 and β-actin for 2 h in TBS-T, followed by the secondary goat anti-rabbit and mouse horseradish peroxidase–immunoglobulin G (diluted 1:5000; Santa Cruz Biotechnology, Santa Cruz, CA, USA). Western blot band densities were quantified using Image Lab software version 6.1 (Bio-Rad, Hercules, CA, USA) and expressed as percentages relative to the control.

### 4.6. Interleukin-6 Immunoassay

Serum IL-6 levels were measured using an IL-6 Immunoassay Kit (R&D Systems, Minneapolis, MN, USA). The IL-6 density of the substrate was measured using a spectrophotometer at excitation and emission wavelengths of 540 and 570 nm, respectively.

### 4.7. Statistical Analysis

Data are presented as mean ± standard deviation. The normality of the variables was evaluated using the Shapiro–Wilk test. An independent sample *t*-test was used to compare the experimental and control groups. SPSS 22.0 (IBM, Armonk, NY, USA) was used for statistical analyses. A *p*-value < 0.05 was considered statistically significant.

## 5. Conclusions

In conclusion, we demonstrated that sepsis-induced ALI was most prominent at 12 h after sepsis in a CLP animal model. Furthermore, sepsis-induced ALI was significantly aggravated in CKD. However, enhanced lung HSP-70 expression, which protects against sepsis-induced ALI, was not observed. This study suggests that sepsis-induced organ damage can become more severe due to improper HSR functioning under the harmful conditions of CKD. These findings may play an important role in future treatment strategies for sepsis in patients with CKD.

## Figures and Tables

**Figure 1 ijms-24-05641-f001:**
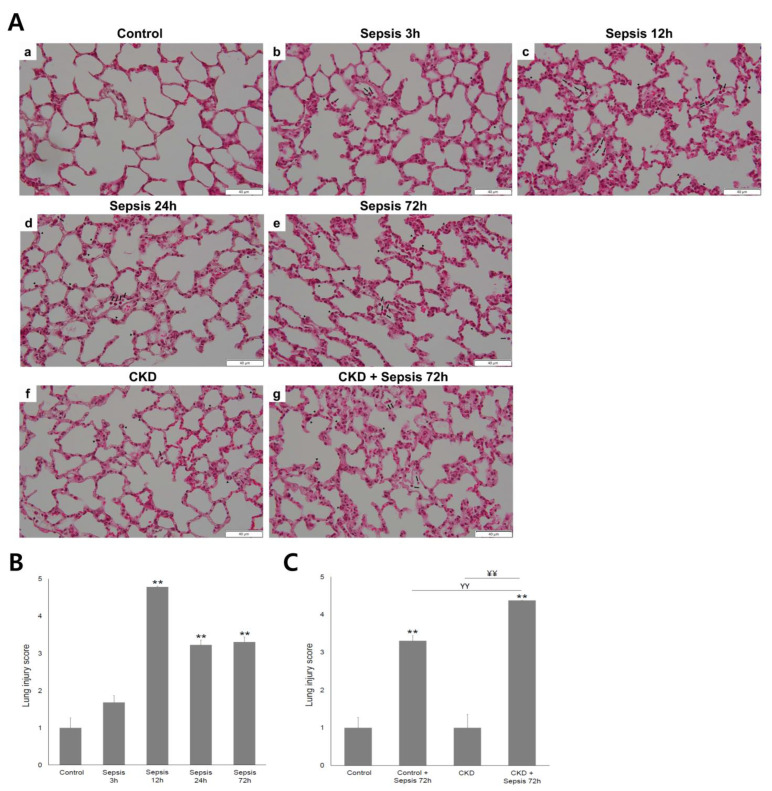
Photomicrographs of lung sections stained with hematoxylin–eosin and changes in lung injury scores according to the course of sepsis: (**A**) in the control group (**a**–**e**), acute lung injury (ALI) was most severe at 12 h of sepsis. Note that neutrophils were identified in the alveolar space (black arrow) and/or the interstitial space (arrowhead). In the chronic kidney disease (CKD) group (**f**,**g**), ALI was aggravated after 72 h of sepsis. Magnification, 400×; (**B**) the lung injury score was highest at 12 h of sepsis in the control group; (**C**) without sepsis, the lung injury score was similar between the control and CKD groups. After 72 h of sepsis, lung injury was significantly more severe in the CKD group than in the control group. Control, sham-operated model; sepsis model with 3, 12, 24, and 72 h after cecal ligation and puncture (CLP); CKD, chronic kidney disease (5/6 nephrectomized) model. ** *p* < 0.01 vs. control; ^ΥΥ^
*p* < 0.01 vs. sepsis 72 h; ^¥¥^
*p* < 0.01 vs. CKD. Error bars represent the standard error of the mean (SEM).

**Figure 2 ijms-24-05641-f002:**
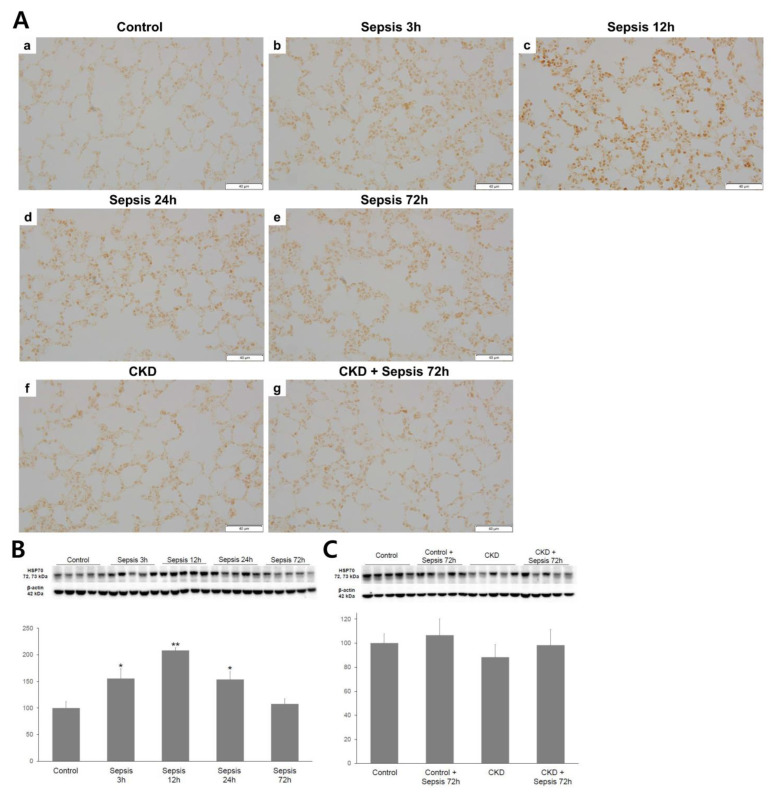
Heat shock protein (HSP)-70 expression in the lung tissue: (**A**) immunohistochemistry of HSP-70 in the lung tissue. Magnification, 400×; (**B**,**C**) Western blot for HSP-70 expression in the lung tissue. Quantification of HSP-70 was standardized based on β-actin expression in the lung: (**B**) in the control group, lung HSP-70 expression was the highest at 12 h of sepsis; (**C**) enhanced lung HSP-70 expression after sepsis induction was not observed in either group. Control, sham-operated model; sepsis model with 3, 12, 24, and 72 h after cecal ligation and puncture (CLP); CKD, chronic kidney disease (5/6 nephrectomized) model. * *p* < 0.05 versus control; ** *p* < 0.01 versus control. Error bars represent the standard error of the mean (SEM).

**Figure 3 ijms-24-05641-f003:**
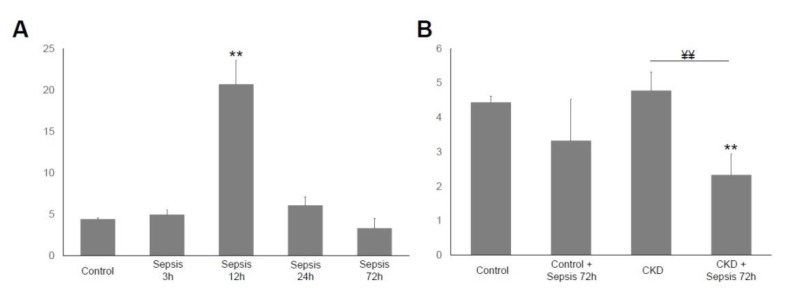
The trend in serum IL-6 concentration level according to the course of sepsis: (**A**) in the control group, serum IL-6 level peaked at 12 h of sepsis; (**B**) Serum IL-6 level was significantly lower in the CKD + sepsis 72 h group than in the CKD group. IL-6, interleukin-6; Control, sham-operated model; sepsis model with 3, 12, 24, and 72 h after cecal ligation and punture (CLP); CKD, chronic kidney disease (5/6 nephrectomized) model. ** *p* < 0.01 vs. control, ^¥¥^
*p* < 0.01 vs. CKD.

**Figure 4 ijms-24-05641-f004:**
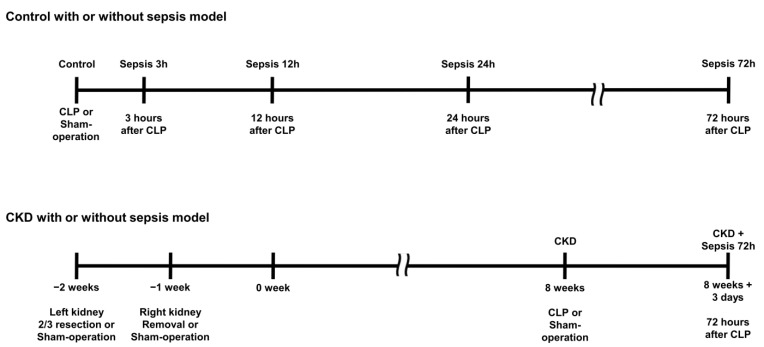
Experimental design. In the control group (sham-operated model), blood and urine collection and lung harvest were performed before cecal ligation and puncture (CLP) and after 3 h, 12 h, 24 h, and 72 h of CLP. In the chronic kidney disease group (CKD, 5/6 nephrectomized), CLP or sham operation was performed after 8 weeks of 5/6 nephrectomy. Blood and urine collection and lung harvest were performed before CLP and after 72 h of CLP.

**Table 1 ijms-24-05641-t001:** Comparison of clinical parameters between the control and chronic kidney disease groups.

	Control	Control with Sepsis at 72 h	CKD	CKD with Sepsis at 72 h
Body weight (g)	520.0 ± 21.3	451.2 ± 51.0 *	425.5 ± 52.4 **	415.2 ± 33.3 **
Urine volume for 24 h (mL)	11.4 ± 1.3	11.6 ± 5.2	30.5 ± 5.2 **^, ΥΥ^	23.4 ± 8.8 **^, Υ^
Urine pH	7.43 ± 0.55	9.16 ± 0.12 **	7.56 ± 0.28 ^ΥΥ^	8.67 ± 0.27 **^, ΥΥ, ¥¥^
Serum Cr (mg/dL)	0.40 ± 0.03	0.54 ± 0.28	0.90 ± 0.26 **	0.90 ± 0.17 **^, Υ^
BUN (mg/dL)	22.9 ± 3.9	25.85 ± 10.7	53.6 ± 14.4 **^, Υ^	37.4 ± 4.5 **^, ¥^
Serum albumin (mg/dL)	2.3 ± 0.2	1.78 ± 02 **	2.1 ± 0.2 ^Υ^	1.9 ± 0.2 *
Cl_Cr_ (mL/min/100 g)	0.37 ± 0.06	0.32 ± 0.20	0.15 ± 0.04 **	0.21 ± 0.03 **^, ¥¥^
Plasma CRP	0.02 ± 0.00	0.02 ± 0.00	0.02 ± 0.00	0.02 ± 0.00

Values are presented as the mean ± standard deviation (SD); Cr, creatinine; BUN, blood urea nitrogen; Cl_Cr_, creatinine clearance; control, sham-operated; sepsis at 72 h, cecal ligation and puncture; CKD, chronic kidney disease model (5/6 nephrectomized); CRP, C-reactive protein; * *p* < 0.05 vs. control, ** *p* < 0.01 vs. control; ^Υ^
*p* < 0.05 vs. Sepsis 72 h, ^ΥΥ^
*p* < 0.01 vs. Sepsis 72 h; ^¥^
*p* < 0.05 vs. CKD, ^¥¥^
*p* < 0.01 vs. CKD.

## Data Availability

Data sharing is not applicable to this article.

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
