# Peer review of "Altered Lung Heat Shock Protein-70 Expression and Severity of Sepsis-Induced Acute Lung Injury in a Chronic Kidney Disease Rat Model"

_ijms, 2023, doi:10.3390/ijms24065641_

Round 1
Reviewer 1 Report
The manuscript by Cho et al. examined examined the relationship between sepsis-induced acute lung injury (ALI) severity and altered lung heat shock protein-70 (HSP-70) expression in chronic kidney disease (CKD). They found that altered lung HSP-70 expression is associated with the worsening of sepsis-induced ALI in patients with CKD. However, the conclusions of this research are not justified by the results. The methodology seems to be correct in most experiments, but the results of this work may not be worth publishing in the current condition. The study requires improvement in many aspects. Please consider the following points:
Major comments:
1. The authors found that altered lung HSP-70 expression was associated with the worsening of sepsis-induced ALI in patients with CKD. However, in the CKD models, the level of HSP-70 had no significant difference in the CKD and CKD + sepsis groups compared to the control group in Figure 2C.
2. The author detected the severity of acute lung injury after sepsis at 3 h, 12 h, 24 h, and 72 h in Figure 1A, however, clinical parameters were not examined after sepsis at 3 h, 12 h, and 24 h in Table 1.
3. In Figure 1A, the results of HE staining should be precisely described in the part of 2.2. Severity of acute lung injury. Lung injury scores had no significant difference in the control and CDK groups as described by the author. However, I believe that the results of HE staining in the CDK group was similar to that in the sepsis group at 24 h, please explain the results.
4. In addition to HSP-70, Other HSPs should be also explored as CDK had no significant effects on the expression of HSP-70.
5. As the author described, lung histology, HSP-70 expression, and inflammatory markers at 3 h, 12 h, and 24 h of sepsis in the CKD group should be further detected.
Minor revision:
1. In Figure 1A, the scale bar should be added in the figures. Enlarged figures should be needed in Figure 1Aa-g to make the neutrophils clearly.
2. In Figure 2A, the scale bar should be added in the figures. Enlarged figures should be needed in Figure 2Aa-g to make the neutrophils clearly.
3. In Figure 2C, the analysis of WB should be further confirmed. According to the results of immunohistochemistry of HSP-70 and WB, the level of HSP-70 in the group of CKD +sepsis 72 h was significantly decreased compared to that in the control group.
Author Response
Major comments:
1. The authors found that altered lung HSP-70 expression was associated with the worsening of sepsis-induced ALI in patients with CKD. However, in the CKD models, the level of HSP-70 had no significant difference in the CKD and CKD + sepsis groups compared to the control group in Figure 2C.
-->First, thank you for your thoughtful review and meticulous comments. In our study, lung HSP-70 expression did not change significantly before sepsis and after sepsis 72h whether it is the control or CKD group. Considering the changing pattern in lung injury score or HSP-70 expression in the control group, we may obtain significant results if we compared outcomes between baseline and sepsis 12 h. As mentioned in the discussion section, we hypothesized that lung injury would be most severe at sepsis 72 h. Contrary to our expectation, lung injury severity, and lung HSP-70 expression were most prominent at sepsis 12 h. Due to several limitations (also mentioned in the discussion section), we could obtain limited data in the CKD group, unlike the control group. In the setting of our experimental study, lung injury recovery was performed at sepsis 72 h, and normalization of lung HSP-70 may be followed. Hence, no significant difference between baseline and sepsis 72 h was observed in the control and CKD groups.
2. The author detected the severity of acute lung injury after sepsis at 3 h, 12 h, 24 h, and 72 h in Figure 1A, however, clinical parameters were not examined after sepsis at 3 h, 12 h, and 24 h in Table 1.
--> We conducted the experiment under the assumption that lung injury would be most severe at sepsis 72 h. In the CKD group, data were obtained at baseline and 72 h of sepsis. Hence, clinical parameters could be compared between baseline and sepsis 72 h. Except for the CRP, other clinical parameters showed that the CKD model was well induced as intended and responses to sepsis was appropriate.
3. In Figure 1A, the results of HE staining should be precisely described in part of 2.2. The severity of acute lung injury. Lung injury scores had no significant difference in the control and CDK groups as described by the author. However, I believe that the results of HE staining in the CDK group was similar to that in the sepsis group at 24 h, please explain the results.
--> We have selected photomicrographs with most prominent histologic changes in each groups. In the CKD group, standard error of the mean lung injury score was larger than other groups. We have changed the photomicrograph that can better reflect the mean lung injury scores.
4. In addition to HSP-70, Other HSPs should be also explored as CDK had no significant effects on the expression of HSP-70.
--> Among many HSPs, we chose the HSP-70 because it is relatively well-studied in an experimental model with lung injury. Of course, other HSPs were not evaluated in the current study due to practical limitations. As you pointed out, we plan to address other HSPs other than HSP-70 in future studies.
5. As the author described, lung histology, HSP-70 expression, and inflammatory markers at 3 h, 12 h, and 24 h of sepsis in the CKD group should be further detected.
--> Thank you for your insightful comment. Undoubtedly, it is a limitation of our study that the CKD group was not divided in detail like the control group to obtain data. In particular, the data at sepsis 12 h will have important meaning. In our further study, these limitations will be supplemented to conduct well-designed studies.
Minor revision:
1. In Figure 1A, the scale bar should be added in the figures. Enlarged figures should be needed in Figure 1Aa-g to make the neutrophils clearly.
--> We have modified and improved the Figure 1 as you suggested.
2. In Figure 2A, the scale bar should be added in the figures. Enlarged figures should be needed in Figure 2Aa-g to make the neutrophils clearly.
--> We have modified and improved the Figure 2 as you suggested.
3. In Figure 2C, the analysis of WB should be further confirmed. According to the results of immunohistochemistry of HSP-70 and WB, the level of HSP-70 in the group of CKD +sepsis 72 h was significantly decreased compared to that in the control group.
--> It was judged to be a problem caused by the low image quality. We recaptured all pictures in Figure 2A in high quality. The magnification was changed from 200 times to 400 times, and the contents were re-described.

Reviewer 2 Report
After careful reviewing of the manuscript (Altered lung heat shock protein-70 expression and severity of sepsis-induced acute lung injury in a chronic kidney disease rat model) i found it interesting and the idea of the study characterized by novelty but i have some questions
1- the plasma CRP level is the same in all groups, is it logic in presence of sepsis?
2- the quality of H&E figure should be improved
3- in figure 1 the H&E plate of CKD and CKD+sepsis nearly the same lung injury score but in plate C you show higher significance not matched with the presented H&E plate
4- Figure 2, the quality of HSP70 immunohistochemistry photos should be improved
Author Response
After careful reviewing of the manuscript (Altered lung heat shock protein-70 expression and severity of sepsis-induced acute lung injury in a chronic kidney disease rat model) i found it interesting and the idea of the study characterized by novelty but i have some questions
1- the plasma CRP level is the same in all groups, is it logic in presence of sepsis?
--> First, thank you for your careful review and comments. We described this point in the discussion section. The intensity of sepsis induction (CLP method: ligation length, number of punctures) seems to be relatively weak to induce severe, irreversible acute lung injury. Moreover, 72 h of the time period may be enough for lung damage repair or other defense mechanisms. For the reasons mentioned above, plasma CRP level was not different in all groups.
2- the quality of H&E figure should be improved
--> We have improved the quality of Figure 1 as you suggested.
3- in figure 1 the H&E plate of CKD and CKD+sepsis nearly the same lung injury score but in plate C you show higher significance not matched with the presented H&E plate
--> We have confirmed that an inappropriate photomicrograph was placed in (f) in figure 1. We have replace it another representative photomicrograph of CKD group was replaced.
4- Figure 2, the quality of HSP70 immunohistochemistry photos should be improved
--> We have improved the quality of HSP-70 IHC photomicrographs in Figure 2.

Round 2
Reviewer 1 Report
no
Author Response
- Provide more accurate definition of sepsis. It is very vaguely mentioned.
--> Thank you for your comments. We modified the sentence about the relationship between sepsis and ALI described in the introduction section to be concise and unambiguous. In addition, the existing references were integrated and attached to the modified sentence.
- As major cause of sepsis is due to lipopolysaccharide (LPS) exposure, please explain in detail.
--> We have inserted the following phrase in the 1st paragraph of the introduction; “Lipopolysaccharide, an essential bacterial cell wall component, induces sepsis as a bacterial endotoxin through a vigorous systemic inflammation response”. Also, we have inserted a reference to the new sentences and reference numbers have been corrected.
- Lines 50- 51: It is incorrect to state that exact mechanism of heat shock response has not been identified. Heat shock response and its regulation as well as its function are rather well understood. Please modify these sentences to be precise.
-->We have changed the sentence as you suggested in the 2nd paragraph of the introduction section.
